# Structural Aspects of the ROS1 Kinase Domain and Oncogenic Mutations

Juliana F. Vilachã [1] 🆔, Tsjerk A. Wassenaar [1,2] 🆔 and Siewert J. Marrink [1,*]

[1] Groningen Biomolecular Sciences and Biotechnology Institute, University of Groningen, 9700 AB Groningen, The Netherlands; j.f.vilacha@rug.nl (J.F.V.); t.a.wassenaar@pl.hanze.nl (T.A.W.)

[2] Knowledge Center Biobased Economy, Hanze University for Applied Sciences, Zernikeplein 11, 9747 AS Groningen, The Netherlands

[*] Correspondence: s.j.marrink@rug.nl

**Abstract:** Protein kinases function as pivotal regulators in biological events, governing essential cellular processes through the transfer of phosphate groups from ATP molecules to substrates. Dysregulation of kinase activity is frequently associated with cancer, ocasionally arising from chromosomal translocation events that relocate genes encoding kinases. Fusion proteins resulting from such events, particularly those involving the proto-oncogene tyrosine-protein kinase ROS (ROS1), manifest as constitutively active kinases, emphasizing their role in oncogenesis. Notably, the chromosomal reallocation of the ros1 gene leads to fusion of proteins with the ROS1 kinase domain, implicated in various cancer types. Despite their prevalence, targeted inhibition of these fusion proteins relies on repurposed kinase inhibitors. This review comprehensively surveys experimentally determined ROS1 structures, emphasizing the pivotal role of X-ray crystallography in providing high-quality insights. We delve into the intricate interactions between ROS1 and kinase inhibitors, shedding light on the structural basis for inhibition. Additionally, we explore point mutations identified in patients, employing molecular modeling to elucidate their structural impact on the ROS1 kinase domain. By integrating structural insights with in vitro and in silico data, this review advances our understanding of ROS1 kinase in cancer, offering potential avenues for targeted therapeutic strategies.

**Keywords:** ROS1; kinase inhibitor; X-ray crystallography; molecular modeling

## 1. Introduction

In cancer, protein kinases can be activated by mutations in nucleotides, the gain or loss of chromosomes (somatic mutations or chromosomal alterations), and gene amplification [1]. The discovery of the echinoderm microtubule-associated protein-like 4 (EML4) and the Anaplastic Lymphoma Kinase (ALK) fusion protein as a druggable target was a benchmark for treating lung adenocarcinomas [2]. This finding highlights the importance of identifying and inhibiting such proteins to successfully treat a subset of non-small cell lung cancer (NSCLC). Later, the proto-oncogene tyrosine-protein kinase ROS (ROS1) (UniProt ID: P08922), an orphan receptor tyrosine kinase, was associated with tumorigenesis and identified in an NSCLC patient with tumor progression. Like ALK, ROS1 can be fused with different partners, the most common being the cell-surface receptor cluster differentiation (CD) 74 [3].

ROS1 and ALK share a high overall sequence similarity (>64%), with this value reaching 84% concerning the ATP binding pocket [4–6]. Type I inhibitors such as crizotinib and lorlatinib have been extensively used in the treatment of these patients. Type II inhibitors have also been studied, especially against mutations [7]. As observed for other kinases, the emergence of kinase point mutations after the use of kinase inhibitors also occurs for ROS1. The classical gatekeeper (L2026M) and solvent front mutations (e.g., G2032R) have been characterized and are associated with a diminished response to type I inhibitors [5,6].

These mutations are often associated with activation of the kinase domain and/or resistance to kinase inhibitors [8]. Understanding the mechanism driving this event can aid the selection of drugs to be prioritized for patient treatment or even lead to the development of novel drugs with a higher affinity for the mutated kinase [9,10]. These mechanisms are driven by structural changes linked to the nature of the mutation [11]. The characterization of the native kinase domain is critical to (I) understand the structural features of this kinase in the bound or apo states and, (II) compare, on a structural level, the overall folding in the presence of mutations. In the case of ROS1, the use of X-ray crystallography was crucial in understanding the binding profile of different kinase inhibitors. However, there is a limitation in four structures available to this day at the Protein Data Bank (PDB). All of them are from the wild-type ROS1 kinase domain; additionally, all are bounded to a small molecule [5,6,12].

In the case of the absence of an experimentally determined structure, the use of molecular modeling can aid in filling this gap. As observed for other kinases, the use of molecular modeling can provide a three-dimensional structure of mutated proteins that are difficult to produce, facilitate the screening of large libraries of compounds, or even the screening of the dynamics of the protein [13]. In our previous review, we provided a broad analysis of the crystal structures of druggable kinases in non-small cell lung cancer (NSCLC) and their fundamental role as templates for modeling studies [14]. In this paper, we extend this analysis to the ROS1 protein and its most common mutations.

## 2. The Native ROS1 Kinase Domain

The fusion product conserves the ROS1 kinase domain, which follows the conserved folding previously mentioned: a bilobal protein with the N- and C-terminal lobes connected by a hinge. As observed from the first ever determined crystal structure of the ROS1 kinase domain, the N-terminal contains a regulatory $\alpha$C-helix and a $\beta$-sheet composed of four strands. The $\alpha$C-helix is connected to the $\beta$3 and $\beta$4 strands. The $\beta$1 and $\beta$2 strands are connected through the G-loop, which acts as the lid of the ATP binding pocket [5]. In the active conformation, the activation loop, a polypeptide region of the C-terminal lobe with increased structural flexibility, is presented in an extended manner with the conserved Aspartic acid–Phenylalanine–Glycine (DFG) motif at its beginning. Once the protein assumes a DFG-in conformation, the DFG phenylalanine side chain is guided into a hydrophobic pocket and engages in hydrophobic stacking interactions [15]. This pocket, explored by Davares et al., contains residues from the $\alpha$C-helix (L2000 and F2004), the DFG motif (F2103), and the other aromatic residues from the C-terminal (A2106 and F2075). A homologous pocket is observed on the ALK-active conformation and is reported to contribute to the maintenance of the active state [4] (Figure 1).

The Protein Data Bank (PDB) contains, to this day, only four available structures of the ROS1 kinase domain (PDB: IDs 3ZBF, 4UXL, 7Z5W, 7Z5X), and all obtained through X-ray crystallography and co-crystallized with different ligands [5,6,12]. From these ligands, the multikinase inhibitor crizotinib was initially designed to inhibit the hepatocyte growth factor receptor (HGFR or c-MET) and later had its application extended to ALK and ROS1. When comparing crizotinib complexes with ALK (PDB: ID 2XP2), ROS1 (PDB: ID 3ZBF), and c-MET (PDB: ID 2WGJ), it is possible to observe a fair overlap of this small molecule in all structures. The drug binds to the ATP binding pocket in all proteins similarly, the 2-amino-pyridine ring engages in two hydrogen bonds with the backbone hinge residues, in the case of ROS1 being E2027 and M2029. Due to the R configuration, the methoxy unit containing the chiral carbon guides the 2,6-dichloro-3-fluorophenyl ring towards the DFG motif at the beginning of the activation loop (Figure 2) [5,6,12].

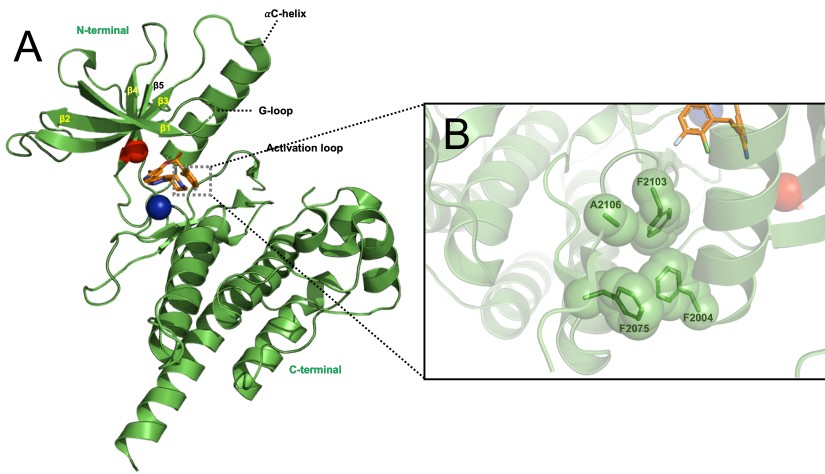

**Figure 1.** (**A**) Schematic representation of active ROS1 kinase domain with crizotinib (PDB: ID 3ZBF). Mutation hotspots, indicated in red (L2026) and blue (G2032), are depicted as spheres. (**B**) The zoomed-in region shows the hydrophobic cluster involving residues L2000, F2004, F2103, A2106, and F2075.

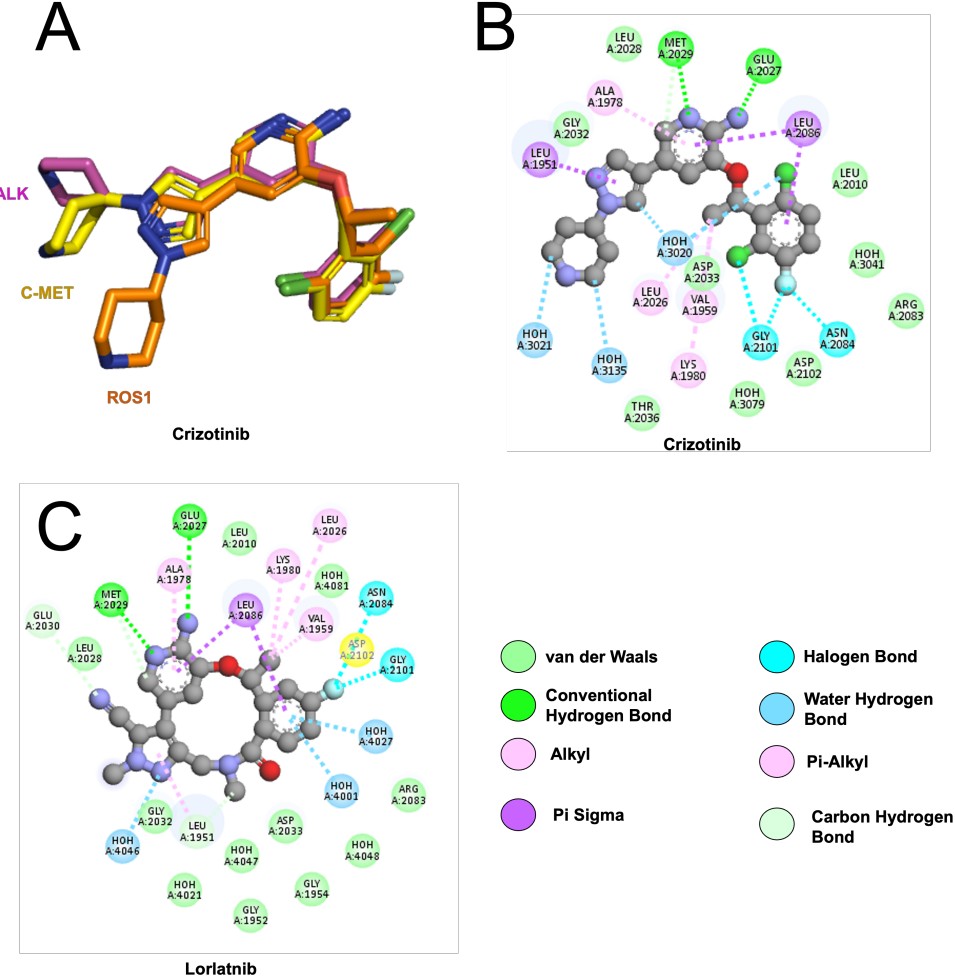

**Figure 2.** (**A**) Overlap of crizotinib extracted from complexes with c-Met (yellow), ALK (magenta), and ROS1 (orange). Two-dimensional representation of first-generation inhibitors, (**B**) crizotinib and (**C**) lorlatinib, with the following color scheme for the atoms: carbon (grey), oxygen (red), nitrogen (blue), chlorine (green), and fluorine (cyan). The interactions were analyzed and plotted using Discovery Studio Visualizer and Pymol.

The major difference between all crystal structures concerns the pyrazole ring; while positioned between two key residues, L1951 and G2032 (ROS1), it retains a level of rotational freedom. Thus, the rotation of the pyrazole ring is associated with the piperidine ring's position. The conformation observed for the pyrazole ring in the ROS1 complex is comparable to the position for the same drug on the ALK receptor (PDB: ID 2XP2), another target for this small molecule. The difference observed for c-MET can be credited to the presence of tyrosine (Y1159) in the place of leucine as observed for ALK and ROS1 (L1198 and L2028, respectively) (Figure 2). The behavior of this five-member ring might affect the binding, as interactions with the G-loop are pivotal for drug binding in type I inhibitors such as crizotinib [16]. In addition, molecular electrostatic potential (MEP) calculations show a concentration of negative charges around the pyrazole and pyridine rings and positive charges in the substituted benzene ring, which could explain the structure–activity relationship of crizotinib and cMET, ALK, and ROS1 targets [17].

Despite the success of crizotinib, there is a limitation in its ability to cross the blood–brain barrier and tackle metastatic incidences of ROS1+ tumors in the brain. As such, crizotinib was modified using a structure-based drug design (SBDD) approach, leading to a macrocyclic final product: lorlatinib. This macrocycle showed a lower propensity for p-glycoprotein (P-gp) efflux and improved blood–brain barrier (BBB) penetration [18]. The design of lorlatinib conserved the hinge-binding amino pyridine scaffold and the chiral center. The fluoro substituent on the aromatic ring was conserved, while the two chlorines were not. The pyrazole ring was N-methylated and substituted at position 3 with a carbonitrile group. The pyrazole ring and the fluoro-phenyl ring were then connected by a methyl-amide motif, leading to the cyclization of the structure [18,19].

As expected, in the co-crystal with native ROS1 (PDB: ID 4UXL), lorlatinib binds in a comparable way to crizotinib. As observed for crizotinib, the aminopyridine core conserves the two hydrogen bonds to the kinase hinge. The fluoro atom polarizes the ortho aryl aromatic C-H groups, leading to electrostatic complementarity with the backbone carbonyl at G2101, a glycine right before the DFG motif that has been explored for drug design [20]. In the crystal structure, the introduced N-methyl amide is close to the G-loop, interacting with the carbonyl group of L1951 and the backbone methylene of G1951. The amide carbonyl from the bridge group interacts with residues K1980 through a water bridge, while in crizotinib, the positioning of the pyrazole ring differs according to the receptor co-crystallization. The pyrazole group is restrained by the cyclization process, which induces the optimal conformation for contact with the C-H of G2032 in a C-H donor–pi interaction [18].

Despite the molecular similarity, lorlatinib and crizotinib presented different inhibition profiles even in the absence of mutations in the kinase domain. Comparison of the atomic displacement parameter, or B-factor, for crystals structures of ROS1$^{WT}$ in complex with crizotinib (PDB: ID 3ZBF) or lorlatinib (PDB: ID 4UXL) provided an overview of how the presence of the ligands can influence the flexibility of key motifs. While for both structures, the highest values for the B-factor were associated with the G-loop (3ZBF: 75 ± 9 (SD); 4UXL: 82 ± 7 (SD)) and the activation loop (3ZBF: 61 ± 16 (SD); 4UXL: 10 ± 17 (SD)), these regions presented a better resolution of the 4UXL crystal structure. The fact that the G-loop region is unresolved for crizotinib indicates a higher flexibility of this motif in the presence of the drug. Overall, analysis of the normalized B-factor of both structures indicated that lorlatinib stabilizes the ROS1 kinase domain more than crizotinib. This effect is especially observed for residues L1951, V1959, and V1979, all located in the N-terminal $\beta$-sheet, which can be associated with the "cooling" of the residues [21].

Approaches relying on docking are often used to identify novel scaffolds; however, they present a low success rate [22]. The low rate of success may be associated with the onerous step of identifying hits within an extensive library, which can be time-consuming and often depends on visual analysis. Altogether, virtual screening frequently provides false positives that do not go further in the development pipeline. Aiming to improve the application of docking in drug discovery, AstraZeneca (AZ) proposed a combination of

structure- and ligand-based drug design (SBDD and LBDD, respectively). Starting from a set of five molecules identified by different approaches, but with considerable ROS1 potency and lack of potency over neurotrophic receptor tyrosine kinase 1 (TrKA) (a kinase anti-target), AZ identified novel ROS1 inhibitor prototypes. Besides consolidating the efficacy of the use of the FastROCS approach and its use in a cloud computing platform, this work also provided two crystal structures of novel ROS1 and ligand complexes [12,23].

One of the novel structures shows the binding of an insulin-like growth factor receptor (IGR-1R) inhibitor (ligand 1) to the ROS1 kinase domain (PDB: ID 7Z5W). In this crystallized structure, the hinge binding motif is the 2-aminothiazole group that engages in two hydrogen bonds with the backbone of M2029 through both nitrogen atoms, the aromatic one acting as an acceptor and the other as a donor. Another hydrogen bond is mediated by a water molecule between the pyrimidine group and the side chain of D2033. The pyrimidine ring from ligand 1 is also positioned between residues L1951 and G2032, comparable to the pyrazole ring in crizotinib. The isoxazole and pyrrolidine rings contribute as linkers to guide the 3-methyl pyrazine ring toward the beginning of the activation loop. In one of the monomers that composes the crystal structure, it is possible to identify an interaction of this motif with the side chain of the D2102 in the DFG motif (Figure 3) [12].

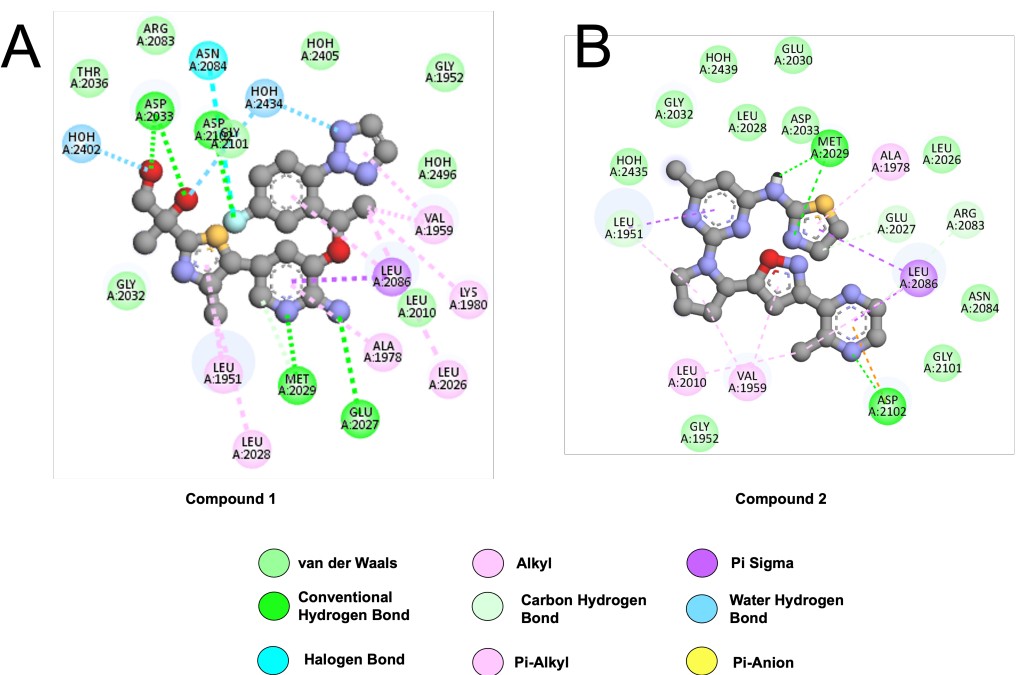

**Figure 3.** Two-dimensional representations of inhibitor prototypes, (**A**) compound 1 (PDB: ID 7Z5X) and (**B**) compound 2 (PDB: ID 7Z5W), with the following color scheme for the atoms: carbon (grey), oxygen (red), nitrogen (blue), sulfur (yellow), and fluorine (cyan). Interactions were analyzed and plotted using Discovery Studio Visualizer.

In parallel, Petrovic also crystallized an ALK inhibitor derived from crizotinib with the ROS1 kinase domain (PDB: ID 7Z5X). This analog conserved the hinge-binding amino pyridine scaffold while modifying the two tails to improve the pharmacokinetic and inhibitory profiles. The piperidine ring was substituted with a 1,2-propanediol group, which interacted with the D2033 residue, and a network of hydrogen bonds was mediated by water molecules. Despite conserving the fluorine atom in the 2-6-dichloro-3-fluoro phenyl ring, one chlorine atom was removed while the other was substituted by a triazole ring. In this crystal structure, the five-membered ring engages in an intramolecular hydrogen bond with the hydroxyl attached to the thiazole ring through a water molecule bridge (Figure 3) [12].

Other ALK inhibitors also showed effective inhibition of ROS1. Entrectinib, a 3-aminoindazole derivative, was initially designed as a type I ALK inhibitor (IC$_{50}$ = 12 nM), with a comparable sensitivity to ROS1 (IC$_{50}$ = 7 nM) [24]. Although a crystal structure of ALK in complex with entrectinib was resolved and is available (PDB: ID 5FTO), a crystal structure of the complex between ROS1 and this small molecule is not available at this moment. However, a study using MD simulations and Molecular Mechanics Poisson–Boltzmann Surface Area (MMPBSA) calculations proposes a mechanism for the binding profile of entrectinib. Overall, the binding affinities obtained from MMPBSA calculations of this drug to ALK and ROS1 are comparable (−40.92 ± 0.32 and −36.60 ± 0.38 kcal/mol, respectively) and both rely on similar residues within the ATP binding pocket, with a major contribution of the L2026 residue [25]. In addition, the major role of L2086 is featured in the stabilization of the ROS1–entrectinib complex due to its consistent interaction with the 3,5-difluoro benzyl ring (Figure 4) [24].

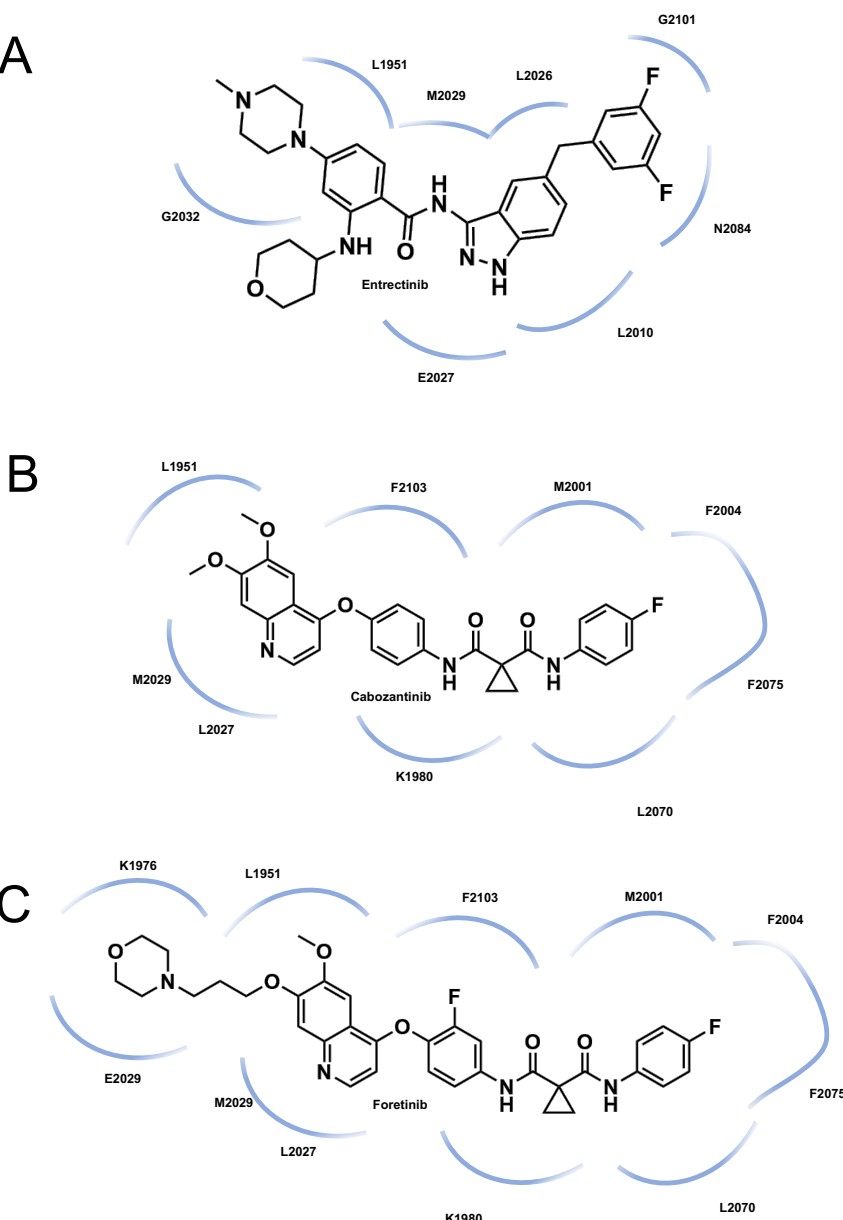

**Figure 4.** Interaction map for (**A**) entrectinib in the ROS1 active conformation proposed by Menich-incheri et al. [24], and (**B**) cabozantinib and (**C**) foretinib in the inactive conformation proposed by Davare et al. [4].

Through homology modeling, using a DFG-out inactive ALK structure (PDB: ID 4FNY), Davare et al. were able to model the inactive conformation of the ROS1 kinase domain [4]. This model showed the absence of the phenylalanine cluster that gave rise to a new pocket: the selectivity pocket. The selectivity pocket is known to be present in the inactive kinase domain and is a less conserved pocket that can be explored for selectivity in kinase drug design [26]. This pocket is located between the αC-helix and the catalytic activation loop, with the latter being dislocated towards the G-loop. Despite being modeled after an ALK inactive structure, the selective pocket of ROS1 is considerably different from the ALK-specific pocket due to the nature of the present amino acids. In this model, the inactive conformation corresponds to the phenylalanine in the DFG motif occupying the ATP binding pocket (DFG-out), and is a suitable target for type II inhibitors such as cabozantinib or foretinib (Figure 5) [4].

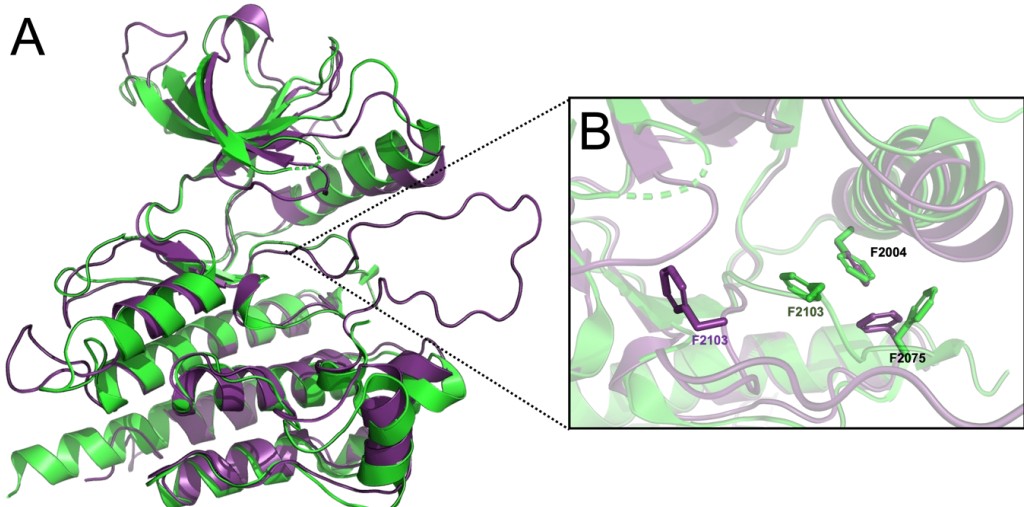

**Figure 5.** (**A**) Superposition of the active (green) experimentally determined (PDB: ID 3ZBF) and the inactive model (purple) obtained through homology modeling (template PDB: ID 4FNY). (**B**) The residue F2103 side chain from the DFG motif is represented as a stick in the active structure (green) and the inactive model (purple).

Analysis of MD simulations of the inactive ROS1$^{WT}$ indicated a more rigid P/G-loop. The same pattern is observed for the activation motif; in the inactive ROS1 model, this motif is more rigid due to its proximity to the P-loop and the αC-helix. The rigidity observed for the αC-helix is due to the proximity of the Q2012 residue to the C-terminal segment of the same helix, restricting its movement toward the catalytic site, which can explain the average pocket volumes for ROS1 and ALK being 186 Å$^3$ and 235 Å$^3$, respectively. Altogether, it was demonstrated that despite the significant sequence homology between ROS1 and ALK, discrepancies in the rigidity and motion of regulatory motifs contribute to the difference observed in the specificity pocket of each kinase and these pockets' volumes [4]. Furthermore, the same inactive ROS1 model was used in relaxed complex scheme (RCS) studies to propose a binding pose for cabozantinib, a type II kinase inhibitor. The proposed binding mode shows the quinoline moiety interacting with the hinge through hydrogen bonds with residue E2027 and M2029, as crizotinib. The aryl linker engages in aromatic stacking with residue F2103 from the DFG motif, while the carboxamide moiety and the K1980 residue interact through a hydrogen bond. The fluorophenyl tail is proposed to be buried in the specificity pocket and interacts with F2004, a residue present in the phenylalanine pocket located at the end of the αC-helix, and F2075 through aromatic stacking interactions. The additional residues M2001, L2070, and I2100 contribute to hydrophobic interactions with this drug (Figure 4) [4].

## 3. ROS1 Point Mutations

### 3.1. L2026M

The same study also proposed a binding pose for foretinib, a potent ROS1 inhibitor structurally related to cabozantinib [7]. The predominant pose obtained from the RCS approach using the apo MD simulations is like the one proposed for cabozantinib with the additional morpholine ring interacting with the K1967 and E2030 residues. A closely related binding conformation was also observed for the same molecule, but with the cMET receptor (PDB: ID 6SD9) [27]. However, a minor cluster of poses obtained from the docking study demonstrated a reversed position of the drug in the binding pocket, with the morpholine closer to the αC-helix. This alternative binding mode highlights a second interaction mode and this may be reason why this mutation retains sensitivity to foretinib even when cabozantinib does not [4].

Gatekeeper residues are in the hinge motif's N-terminal end and are responsible for the accessibility to the "back pocket" of kinases. Gatekeeper mutations are the most common resistance mutation to kinase inhibitors and have been described for many different kinases [28]. In ROS1, the gatekeeper residue is L2026, and the most common mutation is the change of leucine into methionine (Figure 6) [29].

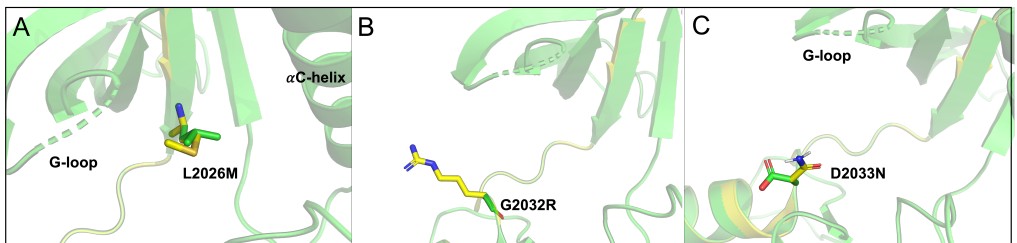

**Figure 6.** Representation of point mutations in the ROS1 kinase domain: (**A**) L2026M, (**B**) G2032R, and (**C**) D2033N. In all figures, the native ROS1 is represented in green with the mutated residues represented in yellow, while being superimposed with the native counter partner residue. Key structural motifs such as the G-loop and the αC-helix are highlighted in the figures to aid identification of their location.

Lorlatinib effectively inhibits BA/F3 cells harboring the CD74-ROS1[L2026M] fusion kinase [6]. While this mutation decreases crizotinib's potency, lorlatinib is efficient in inhibiting the growth of cells expressing CD74-ROS1[L2026M] in a dose-dependent manner [6]. Despite not having an experimentally determined structure for this mutant, earlier studies overlapped the native ROS1 kinase domain to the analogous ALK[L1196M] crystal structure and indicated that the presence of a methionine at position 2026 in the ROS1 protein would likely be accommodated due to lorlatinib lacking the piperidine substituent [6].

### 3.2. G2032R

As observed for the ALK fusions, the treatment of crizotinib in ROS1-positive patients also led to a rise in point mutations in the kinase domain. The first mutation detected was in a lung cancer patient initially treated with crizotinib in a residue exposed to the solvent. The authors reported the G2032R mutation as resistant to crizotinib (Figure 6) [5]. This mutation is analogous to the G1202R ALK mutation that is also linked to crizotinib resistance (Figure 6) [30,31].

Despite there being a crystal structure of the wild-type ROS1 kinase domain complex with crizotinib, attempts to obtain the mutant G2032R mutant were unsuccessful. Thus, homology modeling was performed to analyze the impact of this mutation on the ATP binding pocket. Analysis of the G2032R model indicates that the resistance to crizotinib in the mutant could be caused by the side chain of R2032 clashing with the piperidine ring in crizotinib. It was also proposed that the mutation would not hamper ATP binding, as the ATP concentration required to achieve the half-maximal enzyme velocity was reduced for this mutant compared to non-mutated ROS1 [5].

Later, free energy calculations and MD simulations of the apo ROS1 and the ROS1–crizotinib complex provided a more in-depth view of the effect of this mutation. The analysis of a dihedral defined by the Cα carbon of the residues S1953-A1955-E1958-R2083 showed higher flexibility of the G-loop in the presence of the G2032R mutant than for the ROS1$^{WT}$, thus indicating a more open structure of the G-loop in the apo state. Although similar behavior in the G-loop was observed in the crizotinib-bound state, with a smaller difference between the mutant and non-mutated kinase domain, a comparison of the apo WT and the bounded G2032R dihedral distribution was rather similar. The opened conformation of the G-loop in the mutant was attributed to the inserted arginine, which served as a scaffold for lifting the loop. Despite bringing remarkable insights into the position of the G-loop, the use of classical MD failed to explain the resistance of the mutant to crizotinib [32].

In the same study, using funnel-based well-tempered metadynamics and umbrella sampling to calculate the absolute binding free energy of crizotinib, it was possible to support the correlation between the P-loop rigidity and the effect on the drug binding. The results show that the G2032R mutants sustain a more open conformation of the G-loop, impacting the interaction between the L1951 residue and the drug. This residue in the ROS1$^{WT}$ complex and crizotinib was responsible for the highest contribution to drug binding [32]. Another study using MD simulations of the apo WT and ROS1$^{G2032R}$ supports the overall protein rigidity found in the mutant, especially for the G-loop and the A-loop. The different mobility observed between native and mutant ROS1 was credited to the loss and gain of intramolecular interactions. A stabilizing salt bridge was identified between the guanidinium group of R2032 and the carboxyl in the E1961 in the mutant. Meanwhile, the stabilizing hydrogen bond between E1961 and R1948 identified in the apo ROS1$^{WT}$ simulations was lost in the presence of this mutation. This study showed that not only mutation of the glycine to an arginine leads to steric hindrance of crizotinib binding, but also how the G2032R mutation modifies the intramolecular interactions in the kinase domain that could account for drug resistance [4].

Supporting these findings, MMPBSA calculations showed a decrease in the enthalpic term of free energy calculations. This decrease was especially associated with E2027 and M2029 residues, both involved in hydrogen bonds with the aminopyridine scaffold [29]. Since ROS1 G2032R was proved to be resistant to crizotinib, it was necessary to seek new molecular entities able to inhibit this variant.

In the same study, cabozantinib, an analog to foretinib, showed effective inhibition of the CD74-ROS1$^{G2032R}$ cell line. The docking of cabozantinib in the ensemble of either the native or the mutated ROS1 was considered favorable and mirrored the in vitro assessment [4]. Despite the limited data, the aforementioned description of a probable foretinib binding mode to ROS1 can explain the sustained sensitivity of the G2032R mutant to this type II inhibitor [7].

Despite the G2032R mutation's association with crizotinib resistance, its analog lorlatinib was efficient in the inhibition of the catalytic activity of Ba/F3 cells exhibiting the fusion of CD74 and ROS1$^{G2032R}$ [6]. As previously mentioned, the use of homology modeling associated the resistance of this mutation to crizotinib with a clash between the side chain of the mutated R2032 and the pyrazole and piperidine rings of crizotinib. It was proposed that this clash would be significantly reduced due to the absence of the piperidine ring and the conformational restraint of the pyrazole ring present on lorlatinib [6].

This hypothesis was later supported by MD and free energy calculations. Despite the MM-PBSA/GBSA calculations differing from experimentally determined binding free energies, the magnitude difference observed between the WT and G2032R was replicated. In the mutant, interaction with E2030, M2029, and L1951 residues was weaker, comparable with a similar analysis for crizotinib in the WT binding pocket. In addition, it was shown that despite the mutated R2032 side chain's favorable interaction with lorlatinib, its contribution was comparable to G2032 in the WT protein. As was shown for crizotinib,

the reduced affinity of lorlatinib was associated with an increase in the conformational entropy [33].

### 3.3. D2033N

The aspartic acid at position 2033 is located at the C-terminal of the hinge (Figure 6). ROS1$^{D2033N}$ was initially found in a patient with resistance to crizotinib, which was later confirmed through Ba/F3 cells expressing CD74-ROS1$^{D2033N}$ [34]. Cabozantinib was then administered to the patient, who presented a positive response. The combination of MD simulations and docking studies highlighted the strong electrostatic interaction between the piperidine ring of crizotinib and the negatively charged side chain of D2033. However, the mutant lacks this interaction, as the inserted asparagine lacks the negatively charged side chain. Instead, the N2033 side chain reorients towards the G-loop and engages in a water-mediated hydrogen bond with L1951, leading to a slight loss in flexibility in the G-loop [34].

MD simulations also indicated that the presence of asparagine at position 2033 induced neighboring residue reorientation that would clash with the protonated piperidine ring of crizotinib. Structural alignment of the structural ensemble obtained from MD simulations with the crystal complex of ROS1–crizotinib indicated an electrostatic repulsion between the positively charged nitrogen in crizotinib and N2033. This is not observed for cabozantinib, as the nearest atom of cabozantinib is 5 Å from position 2033 in both the native and mutant [34].

### 3.4. D2113N

The activation loop is not fully characterized in all available crystal structures. This is understandable given the flexible nature of this loop. Interestingly, the exposure of BA/F3 cells expressing native CD74-ROS1 to high concentrations of cabozantinib and foretinib led to the emergence of a D2113N mutation that was resistant to both type II inhibitors. In the inactive ROS1 model used by Davarre, the aspartic acid at position 2113 interacts with R2116 while being repelled by the E2120 in the A-loop [4].

The D2113N mutation was initially described in cell-based resistance screening. This mutation was linked to resistance to type II inhibitors while retaining sensitivity to ROS1/ALK dual inhibitors such as crizotinib, ceritinib, and brigatinib. MD simulations of the inactive conformation of ROS1, native and mutated, indicate the stiffness of the A-loop in the presence of the mutation as the culprit for resistance to type II inhibitors. In addition, the replacement of aspartic acid with an asparagine side chain nullifies the repulsion from the E2120 residue while guiding this residue to form a hydrogen bond with the mutated side chain [4].

As previously mentioned, simulations of the native ROS1 show high occupancy of a salt bridge between the E1997 (αC-helix) and R2107 (A loop) residues, an interaction also described for other kinases in their inactive states [35]. While the ROS1$^{WT}$ lacked such an interaction in active simulations, inactive simulations showed a consistent presence of this salt bridge. However, for the D2113N mutant, the frequency of this interaction was diminished. Analysis of the simulations for the inactive conformation of the ROS1$^{D2133N}$ mutant suggests a displacement of the R2107 residue by the mutated side chain, leading to a loss of the salt bridge. This leads to E1997 side-chain reorientation towards the specificity pocket, partially occupying it and possibly clashing with a type II inhibitor. Interestingly, docking studies mimic the in vitro and MD simulations, showing crizotinib as a more favorable inhibitor than cabozantinib and foretinib [4].

### 3.5. Additional Single and Double Mutations

Cell-based resistance screening performed with Ba/F3 in either native or G2032R CD74-ROS1 also provided additional identification of single or double mutants. Once performed on native CD74-ROS1 cell lines, such studies linked the resistance to cabozantinib with mutations involving F2004 and F2075, residues described as interaction spots for type II

inhibitors [4]. Previous reports had already indicated such residues as possible hotspots for type II resistance due to analogous positions on ALK resistance to kinase inhibitors [36].

Additional point mutations have been described for ROS1, but often with different fusion partners. A more extensive description of single mutations and their associated fusion partners can be read in other papers [37,38]. It is interesting to highlight that most in silico studies focus on the classical G2032R and L206M, thus leaving a gap for other mutations and their impact in the ROS1 kinase domain and further mechanisms of resistance.

Despite compound mutations being commonly found in most of the kinases targeted in clinical studies, up to the moment we finalized this review, no reports of double ROS1 mutations could be found in patient-related material. Resistance screening, exposing cells harboring ROS1 fusion, attempts to predict future double mutations and their response to kinase inhibitors. Treatment of Ba/F3 cells presenting CD74-ROS1$^{G2032R}$ with increasing doses of cabozantinib or foretinib led to a rise in secondary mutations at positions E1974, F2004, I2009, E2020, F2075, N2112, D2113, R2116, M2128, D2143, L2223, and N2224K (Figure 7) [4]. However, the structural impact of these double mutants on ROS1 dynamics and their limited sensitivity to kinase inhibitors remain unknown.

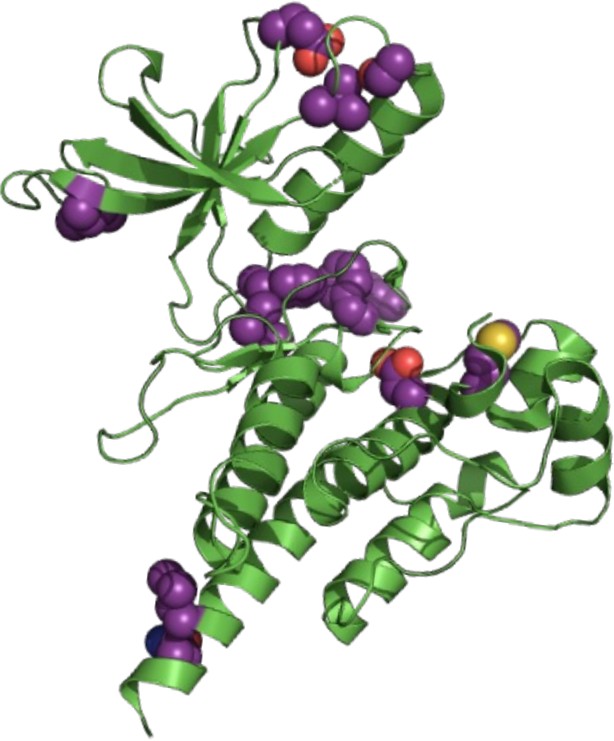

**Figure 7.** Representation of the kinase domain and single mutations. The side chain of native residues with identified mutations is colored in purple and depicted as spheres.

## 4. Conclusions

The discovery of the role of ROS1 fusion proteins in cancer provided a novel druggable target. The high similarity between ROS1 and ALK's ATP binding pockets made the extension of ALK inhibitor use to ROS1 inhibition conceivable. Co-crystallization of crizotinib with the ROS1 kinase domain drew parallels between the binding mode of this small molecule and other kinases. Despite the availability of other crystal structures bearing different ligands, there are no experimentally determined structures of ROS1 mutants.

Mutations in the ROS1 kinase domain are linked to resistance to type I inhibitors and are primarily found in patients treated with crizotinib. The G2032R mutation is in the area exposed to the solvent and has been extensively characterized as a crizotinib-resistant mutation. Molecular modeling of this mutant through different methods shows a clear

impact of the mutant on the overall flexibility of the kinase domain, in addition to possible limitations to crizotinib binding due to its substitutions. Additionally, such an approach explains why ROS1$^{G2032R}$ sustains sensitivity to lorlatinib, a structural analog of crizotinib, despite being resistant to crizotinib.

Gatekeeper mutations are a common class of mutations in kinases. These mutations often render kinase inhibitors ineffective due to an increase in steric clashes in ATP. Reports of crizotinib's interaction with ROS1$^{L2026M}$ were associated with the resistance to entropic changes that were minimized for lorlatinib. However, a deeper understanding of how this mutation affects the kinase domain of ROS1 and the efficacy of other kinase inhibitors is needed. The valuable characterization of the native and mutated (D2113N) inactive conformation of ROS1 provides insights into the binding of type II inhibitors. The contribution of these models and further MD simulations has become more significant due to the lack of an experimentally determined structure of the ROS1 kinase domain in its inactive conformation.

In summary, there is an imperative need to characterize ROS1 and its mutations either by experimental or in silico techniques. Despite the indisputable relevance of experimental data, computational tools have shown the potential to bridge gaps when atomistic details are needed.

**Author Contributions:** Conceptualization, S.J.M. and J.F.V.; writing—original draft preparation, J.F.V.; writing—review and editing, S.J.M., J.F.V. and T.A.W. All authors have read and agreed to the published version of the manuscript.

**Funding:** This research was funded by a bursary from the University of Groningen.

**Conflicts of Interest:** The authors declare no conflicts of interest.

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
