# Peer review of "Structural Aspects of the ROS1 Kinase Domain and Oncogenic Mutations"

_crystals, doi:10.3390/cryst14020106_

Round 1

Reviewer 1 Report

Comments and Suggestions for Authors

Manuscript of Vilacha is well ordered, well illustrated research of structural aspects of the ROS1 kinase domain. Paper is good enough to be published in Crystals and would be interested for readers. I have some small recommendations about clarity improvement, but authors can accept or not. 

Line218.  '...the accessibility to the “back pocket” pocket of kinases.' 

- '"back pocket" of kinases ' seems enough.

Line 218-219. 'A point mutation in this position...' 

But in previous sentence authors are discussing some residues. It is one mutation or several?

Figure 1. Please, contoured at panel A area which is zoomed on panel B.

Figure 7.  I recommend moving labels for residue L1982 and  I2009 to make it more readable

Reviewer 2 Report

Comments and Suggestions for Authors

See attached file

Reviewer 3 Report

Comments and Suggestions for Authors

The manuscript submitted by J. F. Vilacha describes the structures of ROS1 kinase domain with several inhibitors and the oncogenic nature of some of its mutations.

This review is very long and contains a considerable amount of information and it can therefore be interesting for scientists working in this field. However, I must notice that it is very descriptive, handling a publication after the other, and does not draw a general framework where these studies can be inserted.

Specific comments

Line 24, the meaning of NSCLC must be explained. It is done only later, in line 54.

Lines 62-63, the expression “at the beginning and at the end, respectively” might be deleted since it is obvious that it is so.

Line 64, the “activation loop” has not yet been defined.

Line 78, the presence of the R-enantiomer was probably known even before the crystal structure determination.

Figure 1. I would suggest to add the labels “N’ and “C” to the figure.

Caption of Figure 1, the expression “Mutation hotspots are indicated in red (L2026) and blue (G2032) are depicted as sphere” might become “Mutation hotspots, indicated in red (L2026) and blue (G2032), are depicted as sphere.”

Figure 2. The figure B might be rotated by 180 degrees around the vertical axis, so that the molecule is oriented like in A.

Figure 2. In B and C the color code of the atoms is not specified.

Figure 2. The color code on the right includes cases (for example “Pi-Anion”) that are not present on the schemes A and B. These cases should be removed. The same goes in Figure 3.

Line 130, if I remember well, in reference 21 the expression “cooling” is used to indicate that B-factors of the protein loop go down. I liked it and it might be used here, too.

Lines 151-153, this sentence is really unclear.

Line 167, the expression “experimentally determine complex structure” is unclear. It might become “crystal structure of the complex between ROS1 and …”.

Figure 5. The term “alignment” is increasingly used instead of “superposition”. It is a pity, because it is wrong.

Line 189-190, the expression “C-terminal of the same helix” might become “C-terminal segment of the same helix” or “C-terminus of the same helix”.

Line 191, cubic Angstroms are misspelled.

Line 218, the expression ““back pocket” pocket” is really quite poor.

Lines 220-221, the expression “the most common mutation is the mutation of” is really quite poor.

Figure 6. Labels are too small.

Line 233, the expression “(Figure )” might be “(Figure 6)”.

Line 236, “determined” is not necessary

Lines 283-289, these sentences are not clear and should be rephrased.
